# A Quick Simulation Method for Aero-Optical Effects Based on a Density Proxy Model

**DOI:** 10.3390/s23031646

**Published:** 2023-02-02

**Authors:** Bo Yang, He Yu, Chaofan Liu, Xiang Wei, Zichen Fan, Jun Miao

**Affiliations:** 1School of Astronautics, Beihang University, Beijng 100191, China; 2Beijing Institute of Control and Electronic Technology, Beijing 100038, China; 3Qian Xuesen Laboratory of Space Technology, Beijing 100094, China

**Keywords:** celestial navigation, aero-optical effects, density proxy model, optical sensors, hypersonic vehicles, quick simulation

## Abstract

Aero-optical effects caused by high-speed flow fields will interfere with the transmission of starlight, reduce the accuracy of optical sensors, and affect the application of celestial navigation on hypersonic vehicles. At present, the research of aero-optical effects relies heavily on the flow field simulation of computational fluid dynamics (CFD), which requires a great deal of computing resources and time, and cannot satisfy the demand of the rapid analysis of aero-optical effects in the engineering design stage. Therefore, a quick simulation method for aero-optical effects based on a density proxy model (DP-AOQS) is proposed in this paper. A proxy model of the turbulent density field is designed to replace the density field in the CFD simulation, and the proxy model is parametrically calibrated to simulate the optical characteristics of the turbulent boundary layer (TBL) in the external flow field of the optical window. The performance of DP-AOQS in the visible light band is verified from the perspectives of density field distribution, optical path difference (OPD), and fuzzy star map. The simulation results show that the method can quickly provide the distortion results of aero-optical effects in different flight conditions on the premise of ensuring the simulation accuracy. The research in this paper provides a new analytical method for the study of aero-optical effects.

## 1. Introduction

With the rapid development of the aerospace industry in various countries, hypersonic vehicles have become a major research focus of aircraft technology [1,2]. In order to improve the autonomy and accuracy of navigation and guidance, hypersonic vehicles are often equipped with optical imaging detection equipment to provide detection information for navigation and guidance. The application of star sensors in celestial navigation is a particularly extensive research direction at present [3,4]. However, when hypersonic vehicles fly at a high speed in the atmosphere, complex flow structures, such as turbulent boundary layers, shock waves, and expansion waves and mixed layers, may be formed outside the optical window, which makes the density field, temperature field, and velocity field of the air change dramatically [5,6]. When light passes through the complex flow field, the high-speed flow field will disturb the transmission of light and cause optical distortion; that is, aero-optical effects which can make the target image formed by imaging detection equipment appear subject to energy attenuation, offset, blur, jitter, and other phenomena [7,8,9,10]. Therefore, the research of aero-optical effects is of great significance for the application of optical sensors in hypersonic vehicles.

The law of light disturbance caused by high-speed flow field is important to lay a foundation for schemes designed to reduce the influence of aero-optical effects. At present, the main method of research is to obtain the external flow field data of hypersonic vehicles by using high-precision numerical calculation methods, such as computational fluid dynamics (CFD) and direct simulation Monte Carlo (DSMC) [11,12,13]. Then, the optical distortion and image quality evaluation of optical imaging equipment can be obtained through geometric optics, physical optics, statistical optics, and other optical transmission calculation methods [14,15,16]. The focus of those methods is mainly on how to obtain the flow field data, which requires a huge amount of computation and time. When analyzing the influence of flight conditions on the aero-optical effects, most of them adopt the method of repeatedly solving the flow field to calculate the aero-optical effects under the corresponding conditions [17,18,19]. Some scholars have studied the vortex structure of the boundary layer flow field by the CFD method, but the solution of complex differential equation consumes a lot of computational resources [20]. Although the CFD and DSMC methods have a high accuracy in calculating the density field, and the aero-optical effect results calculated from them have a high reliability, the calculation takes a long time and the utilization rate of the data is low. Due to the complexity of turbulence, it is necessary to recalculate the full flow field data when the flight conditions change slightly. Numerical simulation methods such as these, with a high precision and high computation, are used to calculate the density field to analyze the aero-optical effects, which cannot satisfy the requirement of rapidity when conducting aero-optical engineering research at the aircraft design stage. Although some scholars have improved various numerical methods based on hydrodynamics [21], in terms of the aero-optical effects, we mainly focus on the optical properties of the flow field. Therefore, it is necessary to find a quick simulation method for the aero-optical effects expected for the research method based on CFD.

As the calculation of a complex flow field is the main factor affecting computation in aero-optical simulations, a proxy model of the flow field can effectively improve the simulation efficiency of aero-optical effects. Many scholars have verified, through many experiments and numerical simulations, that the aero-optical effects have a corresponding relationship with the boundary layer thickness, Mach number, incoming flow density and other parameters of the optical window boundary layer, and obtained scaling law models such as the root mean square of the optical path difference and jitter angle [22,23,24,25,26]. These high-fidelity aero-optical distortion laws are the basis for matching the flow-field proxy model with the actual high-speed flow field.

The contribution of this paper is to design a boundary layer density proxy model to replace the density field in CFD numerical simulation and realize the rapid simulation of aero-optical effects, as shown in Figure 1. In this paper, the density proxy model is based on the density ellipsoidal vortex model in the aero-optical effects steady-state simulator proposed previously [27]. In contrast to the previous method, this paper considers the density distribution inside the spherical vortex and converts the large-scale vortex structure into the superposition of multiple ellipsoidal vortices. To conform to the real high-speed flow field environment, the motion model of the ellipsoidal vortex structure is added on the basis of the previous static structure to make the density proxy model closer to the real high-speed flow field. In order to ensure the accuracy of the model, polynomial and Bayesian optimization methods are used to calibrate the model. The density and scale control parameters are calibrated based on the boundary layer distortion prediction model. The disturbance parameters solved by the boundary layer linear stability theory are used as the motion parameters of the proxy model. The control parameters of the flow direction movement and dip distribution are calibrated based on the turbulent density fluctuation distribution law. Finally, a density proxy model reflecting the optical characteristics of high-speed flow field is obtained, and the fast simulation of aero-optical effects is completed through the calculation of light.

The remainder of this paper is organized as follows: in Section 2, the design of the density proxy model in DP-AOQS is presented. The calibration method of the control parameters in the density proxy model is described in Section 3. In Section 4, the aero-optical effects in the visible light band are verified from the perspectives of density field distribution, OPD, and fuzzy star map. Finally, the conclusions are drawn in Section 5.

## 2. Design of Density Proxy Model

In this section, the density proxy model of the boundary layer flow field is established. This paper mainly studies the boundary layer flow field outside the optical window, and the thickness of the TBL under hypersonic conditions is about 5–20 mm. In addition, considering the size of the optical window, in order to fully simulate the transmission disturbance of the incident starlight by the TBL flow field outside the optical window fully, a cuboid space above the optical window was selected as the simulation domain with the size of $L_{x}\times L_{y}\times L_{z}=200\;\mathrm{mm}\times 30\;\mathrm{mm}\times 100\;\mathrm{mm}$, as shown in Figure 2. OXYZ is defined as the window coordinate system. The coordinate origin O is located at the centroid of the upper surface of the optical window.

The density sphere vortex model was used to establish the turbulence density proxy model, in which the density sphere vortex model is used to simulate the large-scale structure in turbulence that is much larger than the wavelength of the incident light and has density spatial coherence.

### 2.1. Continuous Density Model of Ellipsoidal Vortex

The density $\rho_{V}(r,t)$ of the large-scale vortex structure is composed of multiple ellipsoidal vortex densities, which can be expressed as follows:(1)$$\rho_{V}(r,t)=\sum_{i}\rho_{i}(r,t)$$
where $\rho_{i}(r,t)$ is the density distribution of the i-th ellipsoidal vortex, which is different from the uniform ellipsoidal vortex in the literature [27]. In order to ensure the continuity of the density field, the internal density ρi(ri,ψi,hi) of the ellipsoidal vortex model is designed as a layered distribution for time t:(2)ρi(ri,ψi,hi)=ρin×ri−Λhi/Λhi2hi2
where ri,ψi and hi are the cylindrical coordinates under the ellipsoidal vortex coordinate system $o_{i}x_{i,V}y_{i,V}z_{i,V}$, as shown in Figure 3.

$o_{i}$ is the centroid of the i-th ellipsoidal vortex; axis $x_{i,V}$ points to the major axis of the ellipsoid and is located in plane OXY; axis $o_{i}y_{i,V}$ points to the minor axis; and Λhi is the major axis length of the ellipsoidal surface at a distance hi from the plane $o_{i}x_{i,V}y_{i,V}$. $\rho_{\mathrm{in}}$ is the maximum spatial fluctuation in the density inside the ellipsoidal vortex, which can be expressed as follows:(3)$$\rho_{\mathrm{in}}={\left(-1\right)}^{N}k_{\rho }F\left(Ma_{\infty },y\right)\Lambda \rho_{\infty }+\rho_{\infty }$$
where $\rho_{\infty }$ is the incoming flow density under the current conditions. ${\left(-1\right)}^{N}$ represents that the density change inside the ellipsoidal vortex may be higher or lower than the incoming flow density. Λ is the major axis length of the ellipsoidal vortex and $k_{\rho }$ is the control parameter of the ellipsoidal vortex density. $F\left(Ma_{\infty },y\right)$ is the Mach number influence factor, which is given by:(4)$$F\left(Ma_{\infty },y\right)=\frac{f\left(y/\delta \right)Ma_{\infty }{}^{2}{\left(1+\frac{\gamma -1}{2}Ma_{\infty }{}^{2}\right)}^{-1/2}}{{\left\{1+\frac{\gamma -1}{2}\left[1-f^{2}\left(y/\delta \right)\right]Ma_{\infty }{}^{2}\right\}}^{3/2}}$$
where $f\left(y/\delta \right)=U_{y/\delta }/U_{\infty }$ is the average flow velocity distribution; $U_{y/\delta }$ is the flow velocity at the height y/δ, which can be approximated by the logarithmic law $c_{1}\ln\left(y/\delta \right)+c_{2}$ in a completely turbulent state; and $U_{\infty }$ is the incoming flow velocity.

### 2.2. Location and Scale Model of Ellipsoidal Vortex

The position and scale distribution of the ellipsoidal vortex structure in the density proxy model are modeled by using the increasing function relationship. The position distribution of the ellipsoidal vortex is established as a Gaussian distribution model about the wall height:(5)$$P\left(\underline{y}\right)=K\exp\left[-\frac{{\left(\underline{y}-{\underline{y}}_{0}\right)}^{2}}{2\sigma^{2}}\right]$$
where $\underline{y}=y/\delta$ and $P\left(\underline{y}\right)$ are the probability of arranging an ellipsoidal vortex at the wall height y. ${\underline{y}}_{0}$ is the wall height with the maximum probability of placing the ellipsoidal vortex. Results from different studies have shown that structures at a height of approximately 0.8δ contribute most to optical distortions [14,28], so we took ${\underline{y}}_{0}=0.8\delta$. σ is the variance of Gaussian distribution, and we took it as a fixed value of 0.5δ and used the size of the gas ellipsoid to adjust the shape of the distorted wavefront. The normalization coefficient K satisfies:(6)$$K\sum_{i=1}^{k_{num}}\exp\left[-\frac{{\left(\underline{y}-{\underline{y}}_{0}\right)}^{2}}{2\sigma^{2}}\right]=1$$
where $k_{num}$ is the total number of ellipsoidal vortices in the simulation domain, and the typical value is between hundreds and thousands [29], so $k_{num}=5000$ was taken in this paper. The size of the large-scale structure in the boundary layer increases at a certain height. The major axis length of the ellipsoidal vortex is described by a quadratic function as follows:(7)$$\Lambda \left(\underline{y}\right)=k_{len}{\left(\underline{y}-{\underline{y}}_{\min}\right)}^{2}+\Lambda_{\min}$$
where ${\underline{y}}_{\min}$ is the wall height of the smallest ellipsoidal vortex; namely, the height for the vertex of the quadratic function, taken as 0.05δ according to the literature [14]. $\Lambda_{\min}$ is the minimum value of $\Lambda \left(\underline{y}\right)$. The range of the minimum density correlation length in TBL is approximately 0.05δ~0.1δ [14,28], so we took $\Lambda_{\min}$ = 0.05δ. $k_{len}$ is taken as the control parameter of the gas-ellipsoidal scale. The density correlation length range of the turbulent boundary layer is about $0.15\delta \leq \Lambda {\left(\underline{y}\right)}_{\max}\leq 0.4\delta$ [17,27]. The constraint range of $k_{len}$ can be obtained according to Equation (7).

### 2.3. Motion Model of Ellipsoidal Vortex

There are disturbance waves with different wave numbers and phase velocities in the supersonic boundary layer [30]. The motion of the ellipsoidal vortex model is modeled as disturbance waves, in which the centroid $\left(x_{i},y_{i},z_{i}\right)$ of the ellipsoidal vortex moves in the form of the superposition of the flow direction traveling waves:(8)$${\dot{x}}_{i}=\sum_{k}{\hat{u}}_{k}\exp\left[j(\alpha_{k}x_{i}+\beta_{k}z_{i}-\omega_{k}t+\varphi_{k})\right]+c.c$$
where ${\hat{u}}_{k}$ is the traveling wave disturbance characteristic function; $\alpha_{k}$ is the k-th order disturbance flow beam; $\beta_{k}$ is the k-th order disturbance spread beam; $\omega_{k}$ is the k-th order disturbance frequency; $\varphi_{k}$ is the k-th order mode phase; and j is an imaginary unit. For the time mode, $\alpha_{k}$ is a real number and $\omega_{k}$ is a complex number; that is, $\omega_{k}=\omega_{k,r}+\omega_{k,j}j$. c.c is a complex conjugate.

In the hypersonic boundary layer, the first mode has an important influence on the transition, and the second mode disturbance becomes the most unstable disturbance wave after the incoming Mach number exceeds four. Therefore, the ellipsoidal vortex center motion is set as the superposition of the first mode and the second mode; that is, k=2. The first mode flow direction beam $\alpha_{1}$ is 0.10892184, the spread direction beam $\beta_{1}$ is ±0.313, the second mode flow direction beam $\alpha_{2}$ is 1.44368594, and $\beta_{2}$ is ±1.5. According to the linear stability theory, the disturbance characteristic functions ${\hat{u}}_{k}$ and $\omega_{k}$ in the control parameters of the ellipsoidal vortex motion can be obtained by solving the Orr–Sommerfeld small-disturbance equation. Considering the directional dependence of the large-scale turbulent structure, the expected inclination of the ellipsoidal vortex was set to 24.17°, and the oblateness was set to 0.447.

## 3. Calibration of Control Parameters in Density Proxy Model

In this section, the control parameters of the density proxy model are calibrated. $\left\{k_{\rho },k_{len}\right\}$ was used to control the position distribution of the ellipsoidal vortex and the spatial fluctuation of the density in the density proxy model, which can be calibrated by the methods of polynomial and Bayesian optimization.

### 3.1. Density and Scale Control Parameter Constraints

The density control parameter $k_{\rho }$ affects the density spatial fluctuation of the ellipsoidal vortex structure, and the scale control parameter $k_{len}$ affects the size distribution of the ellipsoidal vortex, both of which are closely related to the incoming flow state. The density correlation length range of the turbulent boundary layer is about 0.15δ~0.4δ [17,27]. The constraint range of $k_{len}$ can be obtained according to Equation (7). To determine the constraint range of $k_{\rho }$, the Strehl Ratio (*SR*) of the density proxy model was calculated in a steady state for different control parameters under the condition of a flight altitude of 10 km and Mach number of 5 Ma, as shown in Figure 4.

It can be seen from Figure 4 that $SR_{s}$ gradually decreases from 1 to 0 with the increase in the control parameters $k_{\rho }$ and $k_{len}$. When $k_{\rho }\leq 0.001$, the change in $SR_{s}$ with the increase in $k_{len}$ can be ignored, and the turbulence density proxy model cannot reflect the spatial fluctuation of turbulence. When $k_{\rho }\geq 100$, $SR_{s}$ decreases rapidly with the increase in $k_{len}$ and the decreasing speed continues to increase with the increase in $k_{\rho }$, resulting in the excessive spatial pulsation of the turbulence proxy model. Therefore, the upper limit $k_{\rho }^{b}$ and lower limit $k_{\rho }^{a}$ are used for the density control parameter $k_{\rho }$, meeting the following conditions:(9)$$\left\{\begin{array}{c} k_{\rho }^{a}=\arg\underset{k_{\rho }}{\max}\;SR_{s}(k_{\rho },k_{len}^{b})\geq SR^{a}\\ k_{\rho }^{b}=\arg\underset{k_{\rho }}{\min}\;SR_{s}(k_{\rho },k_{len}^{a})\leq SR^{b},k_{\rho }\geq k_{\rho }^{a} \end{array}\right.$$
where $SR^{a}$ and $SR^{b}$ are the minimum and maximum values of *SR* under the working conditions to be calibrated, respectively.

### 3.2. Calibration of Density and Scale Control Parameters

Under the given inflow conditions, the mean square value of OPD of the TBL can be described by the boundary layer distortion prediction model [25]:(10)$${\mathrm{OPD}}_{rms}^{}=BK_{GD}\rho_{\infty }Ma_{\infty }^{2}\delta \sqrt{C_{f}}F\left(Ma_{\infty }^{}\right)$$
where B is a constant, taken as 0.2, $F\left(Ma_{\infty }^{}\right)$ is an empirical function of the incoming Mach number, and $C_{f}$ is the wall friction coefficient of the TBL of the flat plate, which is related to the Reynolds number $Re_{x}$. The density fluctuation and scale structure in turbulence are affected by the Mach number and Reynolds number of the incoming flow. The density control parameters affect the density fluctuation of the ellipsoidal vortex structure, and the scale control parameters affect the size distribution of the ellipsoidal vortex, both of which determine the optical characteristics of the density proxy model. Therefore, the control parameters $k_{\rho }$ and $k_{len}$ can be calibrated as a function of $Ma_{\infty }^{},Re_{x}$, and flight altitude H, which can be expressed as follows:(11)$$\begin{array}{c} k_{len}=g_{len}(Ma_{\infty },Re_{x},H)\\ k_{\rho }=g_{\rho }(Ma_{\infty },Re_{x},H) \end{array}$$

The working condition is expressed as $y={\left[\begin{array}{ccc} Ma_{\infty } & Re_{x} & H \end{array}\right]}^{T}$.$g_{len}(Ma_{\infty },Re_{x},H)$ and $g_{\rho }(Ma_{\infty },Re_{x},H)$ are expressed as g(y). With the boundary layer distortion prediction model as the constraint condition, the optimal function mapping $g^{*}$ is obtained; that is, a variational problem is solved:(12)$$g^{*}=\arg\underset{g}{\min}{\left\|OPD_{rms}^{p}(y)-OPD_{rms}^{s}(g(y))\right\|}^{2}{}_{2}$$
where $OPD_{rms}^{p}(y)$ is the root mean square of the OPD of the boundary layer distortion prediction model under the given working condition y, and $OPD_{rms}^{s}(g(y))$ is the root mean square of the OPD of the density proxy model. Because the gradient information of the control parameter output by the density proxy model cannot be obtained, the traditional variational method cannot be used to solve the variational problem in Equation (12). We transformed it into a parameter optimization problem, which is solved by a gradient-free parameter optimization algorithm.

*SR* in a density proxy model and distortion prediction model can be described as the form of multiplication or the addition of two univariate continuous functions with variable separation. Considering the existence of the multiplication form of the Mach number and Reynolds number in the distortion prediction model, we conducted research in the form of multiplication:(13)$$SR=h_{1}(x_{1})\cdot h_{2}(x_{2})$$
where $x_{1}$ and $x_{2}$ in the density proxy model are $k_{len}$ and $k_{\rho }$, respectively, and $h_{1}(x_{1})$ and $h_{2}(x_{2})$ are the monotonic increasing functions. $x_{1}$ and $x_{2}$ in the distortion prediction model are $Ma_{\infty }^{}$ and $Re_{x}$, respectively, $h_{2}(x_{2})$ is the monotonic decreasing function, and the monotonicity of $h_{1}(x_{1})$ is uncertain. Therefore, the control parameters $k_{len}$ and $k_{\rho }$ can be calibrated as the multiplication of two continuous functions with one variable separated:(14)$$\begin{array}{c} k_{len}=g_{len}{}_{,1}(Ma_{\infty })\cdot g_{len}{}_{,2}(Re_{x})\\ k_{\rho }=g_{\rho ,1}(Ma_{\infty })\cdot g_{\rho ,2}(Re_{x}) \end{array}$$

The univariate continuous functions $g_{len}{}_{,i}$ and $g_{\rho ,i}$ can be fitted by the power function polynomial. The polynomial fitting function is set as g(x;a,N), where N is the highest order term of the power function and $a={\left[\begin{array}{cccc} a_{0} & a_{1} & \cdots & a_{N} \end{array}\right]}^{T}$ is the coefficient vector of the power function.
(15)$$g(x;\pi ,N)=\sum_{k=0}^{N}a_{k}x^{k}$$

The coefficient $a_{1},a_{2},a_{3},a_{4}$ of the power function polynomial and the highest degree term $N_{1},N_{2},N_{3},N_{4}$ of the power function polynomial are parameters to be calibrated. For the convenience of writing, A and N are represented as $A=\left[\begin{array}{cccc} a_{1} & a_{2} & a_{3} & a_{4} \end{array}\right]$ and $N={\left[\begin{array}{cccc} N_{1} & N_{2} & N_{3} & N_{4} \end{array}\right]}^{T}$, respectively. Then, Equation (11) can be written as:(16)$$\begin{array}{c} k_{len}=g(Ma_{\infty };a_{1},N_{1})\cdot g(log(Re_{x});a_{2},N_{2})\\ k_{\rho }=g(Ma_{\infty };a_{3},N_{3})\cdot g(log(Re_{x});a_{4},N_{4}) \end{array}$$

The surface shape of $SR_{p}$ distribution with $Ma_{\infty }^{}$ and $Re_{x}$ in the distortion prediction model changes obviously with the height. When fitting the functional relationship between $\left\{k_{\rho },k_{len}\right\}$ and $\left\{Ma_{\infty }^{},Re_{x}\right\}$ by polynomial, it is necessary to calibrate the power function coefficient matrix A and the highest-order term N by height H. In order to determine the range of coefficient A to be optimized, the input variable $Ma_{\infty }^{},Re_{x}$, and the output variable $k_{\rho },k_{len}$ are normalized, and the value range is limited to the value interval [0,1]. $Ma_{\infty }^{}$ and $log(Re_{x})$ are calibrated according to the range of working conditions. Considering the scope of application of the distortion prediction model, the value range of $Ma_{\infty }^{}$ is limited to $\left[1.5,5\right]$; $Re_{x}$ is limited to $\left[10^{6},10^{8}\right]$; $k_{len}$ is limited to $\left[k_{len}^{a}/\gamma_{R},k_{len}^{b}/\gamma_{R}\right]$; and $k_{\rho }$ is limited to $\left[k_{\rho }^{a}/\gamma_{R},k_{\rho }^{b}/\gamma_{R}\right]$. $\gamma_{R}$ is a trust coefficient which is used to change the range of the control parameters to be calibrated. Additionally, $\gamma_{R}$ is considered as a hyper-parameter, and the value range is set to $\left[0.5,2\right]$. When $\gamma_{R}<1$, the calibration range of the control parameters is reduced, and when $\gamma_{R}>1$, the calibration range of the control parameters becomes larger.

After the normalization of the input variables $Ma_{\infty }^{},log(Re_{x})$ and output variables $k_{len},k_{\rho }$, the value of the higher-order term $a_{ij}$ of the power function polynomial coefficient is naturally limited to $\left[-1,1\right]$, and the range of the zero-order power function coefficient A is $\left[0,1\right]$. Thus, coefficient matrix A satisfies the following equation:(17)$$\begin{array}{c} A^{a}\leq A\leq A^{b}\\ A^{a}=\left[\begin{array}{cccc} 0 & 0 & 0 & 0\\ 0 & -1 & 0 & -1\\ -1 & -1 & -1 & -1\\ -1 & -1 & -1 & -1 \end{array}\right],A^{b}=\left[\begin{array}{cccc} 1 & 1 & 1 & 1\\ 1 & 0 & 1 & 0\\ 1 & 1 & 1 & 1\\ 1 & 1 & 1 & 1 \end{array}\right] \end{array}$$

The minimum value of the highest power term of the constrained power function is 3 and the maximum value is 20. The parameters A and N are calibrated by iterative optimization. For the fixed hyper-parameters N and $\gamma_{R}$ at the given flight altitude $H_{0}$, the parameter A is optimized and solved. With the boundary layer distortion prediction model as the constraint, the optimal solution $A^{*}$ of the parameter A to be calibrated is an optimization problem:(18)$$\begin{array}{l} A^{*}=\arg\underset{A}{\min}\sum_{M,Re_{x}}{\left\|OPD_{rms}^{p}(Ma_{\infty },Re_{x})\left|{}_{H_{0}}\right.-OPD_{rms}^{s}(Ma_{\infty },Re_{x};A)\left|{}_{H_{0}}\right.\right\|}^{2}{}_{2}\\ \;\;\;\;\;subject\;to\;A^{a}\leq A\leq A^{b} \end{array}$$
where $OPD_{rms}^{s}(Ma_{\infty },Re_{x};A)$ is the root mean square of the simulated OPD of the density proxy model at flight altitude $H_{0}$. For a given flight altitude $H_{0}$, the parameter N and trust coefficient $\gamma_{R}$ can be regarded as the hyper-parameters for optimization, and the loss function is defined as:(19)$$\begin{array}{c} Loss\overset{def}{=}\underset{A}{\min}\sum_{M,Re_{x}}{\left\|OPD_{rms}^{p}(Ma_{\infty },Re_{x})\left|{}_{H_{0}}\right.-OPD_{rms}^{s}(Ma_{\infty },Re_{x};A)\left|{}_{H_{0}}\right.\right\|}^{2}{}_{2}\\ \;\;\;\;\;+a_{P}\log(n_{iter}) \end{array}$$
where $n_{iter}$ is the total number of iterations in the optimization, $a_{P}\log(n_{iter})$ is the penalty function of the total number of iterations, and $a_{P}$ is the penalty coefficient, taken as 0.001. When the number of iterations is too large, it means that the iterative optimization effect of the polynomial parameters is poor, and the polynomial form is not suitable as a fitting function. For the above hyper-parameter optimization problem, the Bayesian optimization algorithm is used to solve it:(20)$$x^{*}=\underset{x\in Χ}{\arg\max}f(x)$$
where x is the parameter to be optimized, and its value range is Χ. f(x) is the objective function, and its gradient information is unknown. The basic idea of Bayesian optimization is to estimate the posterior distribution of the objective function based on the Bayesian principle. It is usually necessary to make assumptions about the model of the posterior distribution, estimate the posterior distribution on the basis of the observed data, and select new sampling points by maximizing the effect function to achieve the posterior distribution approaching the true distribution [31].

For $H_{0}=15\;\mathrm{km}$, the solution results of the hyper-parameters based on Bayesian optimization are shown in Figure 5, where (a) and (b) are the iterative scatter distribution of the hyper-parameter N. The red point is the optimal value, and the blue point is the initial iteration point. The size of the point increases with the number of iterations. Figure 5c shows the iteration trajectory of the trust coefficient $\gamma_{R}$, which reaches the optimal value after 30 iterations. The change curve of the loss function Loss is shown in Figure 5d. The prismatic sign is the loss function observed at the sampling point in the current iteration. The circle sign is the minimum value of the loss function in the current iteration. When the number of iterations is 12, the minimum value of the loss function is close to convergence. When the minimum value of the loss function reaches the tolerance range in 30 iterations, the Bayesian optimization solution is completed.

$SR^{s}$ of the turbulence density proxy model and $SR^{p}$ of the distortion prediction model after calibration are shown in Figure 6a,b. After polynomial calibration, the fitting error $\left|SR^{s}-SR^{p}\right|$ of *SR* is within $2\times 10^{-3}$.

## 4. Simulation and Analysis

In this section, with a star sensor as the object, the simulation performance of DP-AOQS is verified from the characteristics of the density field distribution, OPD, and fuzzy star map of the density proxy model.

### 4.1. Verification of Optical Characteristics of Density Proxy Model

For the simulation conditions of $H=15\;\mathrm{km},Ma_{\infty }=3,Re_{x}=10^{6}$, the density fluctuation, and OPD of the density proxy model in the XY plane after being calibrated in Section 3 are shown in Figure 7. It can be seen from Figure 7 that the OPD fluctuated greatly with the flow direction distance because there were ellipsoidal vortices with uneven density fluctuations in the flow direction. For example, at t=5 ms, there were wave peaks in the OPD near x=−0.08 m, and there were many ellipsoidal vortices (near y=0.02 m−0.03 m) with positive density fluctuations near x=−0.08 m. With the passage of time, the movement of the vortex structure in the direction of flow made the distribution of OPD fluctuate in both the time and space dimensions, which is consistent with the pulsation characteristics of the turbulent boundary layer [32]. It only took 200 s for DP-AOQS to obtain a set of results (simulation environment: Matlab R2021b, Intel i9-9900X CPU, 128 G RAM). A high-precision flow field structure calculated by CFD required 10–20 h (simulation environment: a high-performance server with AMD EPYC ROME 7H12, 128-cores CPU, 256 G RAM, the number of grids was more than 6 million, and the generation of large eddy simulation (LES) was more than 20,000).

In order to compare with the OPD calculated by the CFD, the OPD and projection surface of the density proxy model were obtained under the condition $H=20\;\mathrm{km},Ma_{\infty }=3.8$, as shown in Figure 8a. The OPD of the flow field near the concave window under the same simulation conditions by the CFD is shown in Figure 8b [19]. It can be seen that there were also many peaks and troughs in the optical path difference surface. Figure 8 demonstrates that DP-AOQS can reflect the variation trend of the optical properties of the actual turbulent boundary layer flow field.

The flow direction OPD caused by the density fluctuation of the turbulent structure has certain amplitude frequency characteristics. Through experiments and similarity analysis, the amplitude spectrum model of the flow direction OPD was determined as follows [25]:(21)$${\mathrm{OPD}}_{x}\left(St_{\delta x}\right)={\hat{\theta }}_{peak}\frac{1}{1+{\left(St_{\delta x}/0.75\right)}^{5/3}}Ma_{\infty }^{2}f_{s}\frac{\rho_{\infty }}{\rho_{SL}}{\left(\frac{\delta }{U_{\infty }}\right)}^{2}$$
where $St_{\delta x}$ is the normalized frequency of flow direction, satisfying $St_{\delta x}=\delta f_{s}/U_{\infty }$. $f_{s}$ is the sampling frequency, $\rho_{SL}$ is the reference density of the sea level, $\rho_{\infty }$ and $U_{\infty }$ are the incoming flow density and velocity, respectively, and ${\hat{\theta }}_{peak}$ is the peak value of the amplitude. To eliminate the difference caused by working conditions, we normalized Equation (21) to obtain the flow direction normalized spectrum ${\hat{\mathrm{OPD}}}_{x}\left(St_{\delta x}\right)$, as shown in the following equation:(22)$${\hat{\mathrm{OPD}}}_{x}\left(St_{\delta x}\right)=\frac{{\mathrm{OPD}}_{x}\left(St_{\delta x}\right)}{\max({\mathrm{OPD}}_{x}\left(St_{\delta x}\right))}=\frac{1}{1+{\left(St_{\delta x}/0.75\right)}^{5/3}}$$

The amplitude spectrum model, density proxy model, and CFD flow field in the literature [19] were used to calculate the OPD spectrum distribution, as shown in Figure 9. It can be seen from Figure 9 that the density proxy model in this paper had a downward trend as the flow direction frequency increased, which is consistent with the trend of the amplitude spectrum model and the OPD spectrum obtained from the CFD calculation. However, the amplitude of the OPD spectrum in this paper decreased rapidly in the interval $St_{\delta x}=10^{-1}-10^{0}$ because the flow direction motion model only used the first mode and the second mode for superposition, while there are many higher-order mode disturbance waves in the boundary layer. It is normal for the OPD spectrum to show buffeting in the interval $St_{\delta x}=10^{0}-10^{1}$, which is caused by spatial discrete sampling, and amplitude mutation will occur at uncovered frequency points.

### 4.2. Monte Carlo Simulation under Different Working Conditions

The OPD of the density proxy model at 10 km and 15 km was verified by the Monte Carlo method, where the condition was $Ma_{\infty }\in \left[2,5\right],Re_{x}\in \left[1\times 10^{6},1\times 10^{8}\right]$, and the flight altitude was $H\in \left[H_{0}-1,H_{0}+1\right]\mathrm{km}$, where $H_{0}$ is 10 km or 15 km. The results of the Monte Carlo verification are shown in Figure 10, where there were 100 sampling points in total.

From the relative error distribution of OPD in Figure 10, the fitting error of the density proxy model was within 10% at 15 km, while the flight altitude was 10 km, and the fitting error was within 12%, satisfying the accuracy requirements of the simulation. When the altitude decreases, the fitting error increases. This is because when the height decreases, the surface of *SR* will bend more violently with {Ma∞,Rex}, and the error of the control parameters at the uncalibrated height increases. In general, the calibration of the control parameters of the density proxy model at a given altitude within 1 km could satisfy the accuracy requirements of OPD, and the calibration accuracy was higher than the steady-state simulation method proposed in the literature [27].

### 4.3. Simulation of Distorted Star Maps

After the starlight passes through the high-speed flow field, the star map on the image plane of the star sensor will degenerate. Because the distance between the star and the aircraft observed by the star sensor is far greater than the distance between the Earth and the sun, the observed star is regarded as a point light source at infinity. The star point signal on the CCD image plane is concentrated in a small circular area, where the energy distribution can be assumed to be the Gaussian distribution. The light field distribution on the image plane is the gray distribution of the diffraction image of a single star, and the display gray level of the corresponding pixel is obtained after gray level aggregation.

Under the condition of $Ma_{\infty }=3,Re_{x}=1\times 10^{6}$, the simulation results of the two distorted stars at 15 km and 5 km were as shown in Figure 11. In the simulation, the focal length of the star sensor was f=61.88 mm, the pupil size was 50 mm, the pixel size was 15μm, the field angle was $FOV=20^{\circ }$, and the number of pixels on the image plane was 1456×1456.

The gray value at the center of the distorted star map is listed in Table 1. When the flight altitude was 15 km, the $SR_{p}$ of the distortion prediction model was 0.8613, and the $SR_{s}$ of the turbulence density proxy model was 0.8650. At this point, compared with the standard star map, the gray distribution of star A and star B in the distorted star map was still the center point, and the energy was mainly concentrated in the range of one pixel in each direction from the center point. The gray value of the center point in the distorted star map decreased by 5–9 gray values.

When the flight altitude was 5 km, the $SR_{p}$ of the distortion prediction model was 0.0621, and the $SR_{s}$ of the turbulence density proxy model was 0.0645. Compared with the standard star map, the brightest point is still the center point of stars A and B in the distorted star map, but the energy distribution has spread in all directions, spreading 1–2 pixels. The gray value of the center point in the distorted star maps is greatly reduced, and the gray value of star A is the most obvious, falling 79 gray values. The distorted star maps obtained by DP-AOQS can be used for a reference in the design stage of celestial navigation in hypersonic vehicles. In addition, in order to highlight the advantages of this study, the distorted star map results of this paper are compared with the previous research methods. The star map obtained from the previous research [27] is shown in Figure 12.

By comparing the star map results in Figure 12 with those in Figure 11c,d, it can be found that the star map results are in line with objective laws, and the diffusion effect of the star map is more obvious because of the density change in the interior of the ellipsoid vortex and the motion model of the ellipsoid vortex considered in this paper.

## 5. Conclusions

In this paper, a quick simulation method of the aero-optical effects based on the density proxy model (DP-AOQS) is proposed. The density proxy model, considering the optical characteristics of the turbulent steady state and motion state, is designed to replace the density field in CFD numerical simulation. Based on the aero-optical laws obtained from the experiments and simulation analysis, the density proxy model is calibrated by combining polynomial fitting and Bayesian optimization so that the proxy model can reflect the optical characteristics in the actual high-speed flow field. The performance of the aero-optical effects in the visible light band is verified from the perspectives of the density field distribution, OPD, and fuzzy star map. Compared with the traditional CFD simulation method, DP-AOQS can quickly provide the distortion results of aero-optical effects under different flight conditions. It provides a powerful tool for aero-optical effects analysis in the design stage of celestial navigation in hypersonic vehicles.

## Figures and Tables

**Figure 1 sensors-23-01646-f001:**
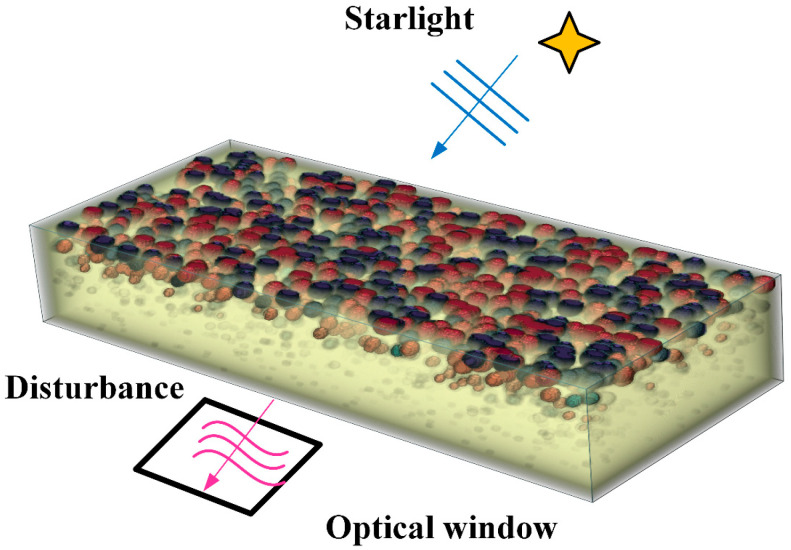
Schematic diagram of boundary layer density proxy model.

**Figure 2 sensors-23-01646-f002:**
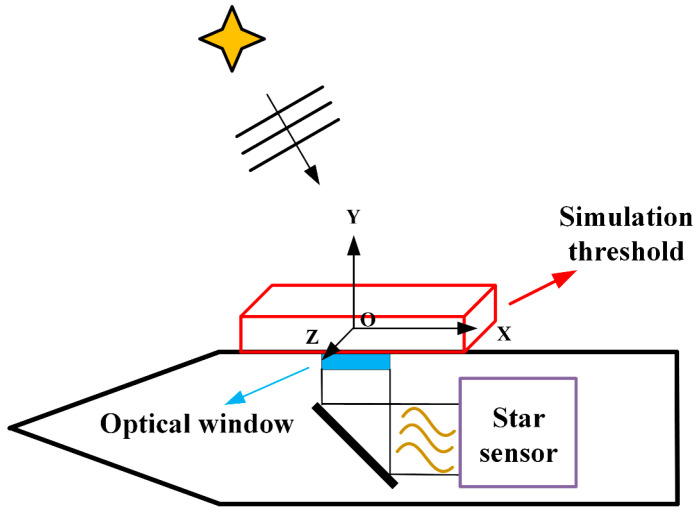
Schematic diagram of simulation threshold of density proxy model.

**Figure 3 sensors-23-01646-f003:**
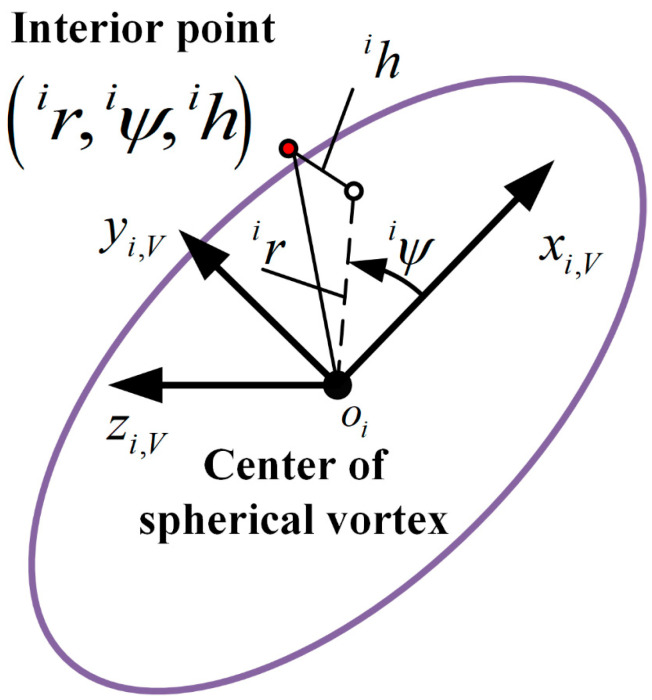
Schematic diagram of ellipsoidal vortex coordinate system.

**Figure 4 sensors-23-01646-f004:**
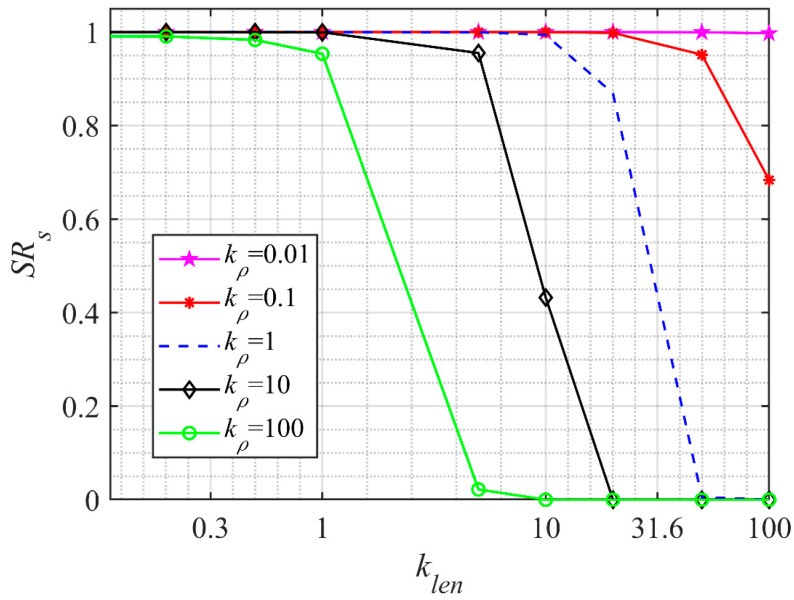
*SR* of density proxy model changing with control parameters.

**Figure 5 sensors-23-01646-f005:**
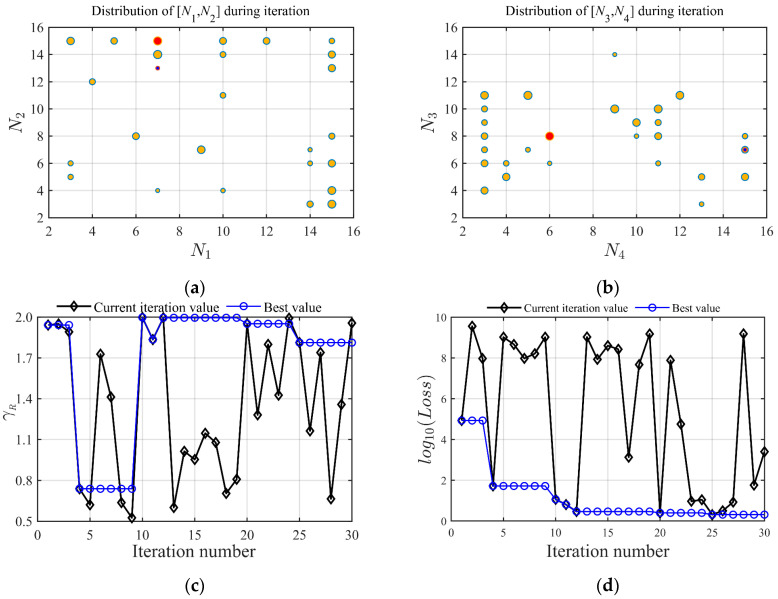
The solution results of the hyper-parameters based on Bayesian optimization: (**a**) iterative scatter distribution of *N*_1_, *N*_2_; (**b**) iterative scatter distribution of *N*_3_, *N*_4_; (**c**) iteration trajectory of the trust coefficient $\gamma_{R}$; (**d**) the change curve of loss function.

**Figure 6 sensors-23-01646-f006:**
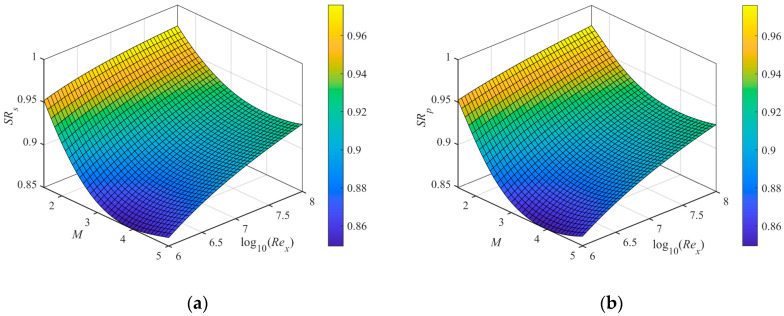
(**a**) $SR^{s}$ of the turbulence density proxy model; (**b**) $SR^{p}$ of the distortion prediction model.

**Figure 7 sensors-23-01646-f007:**
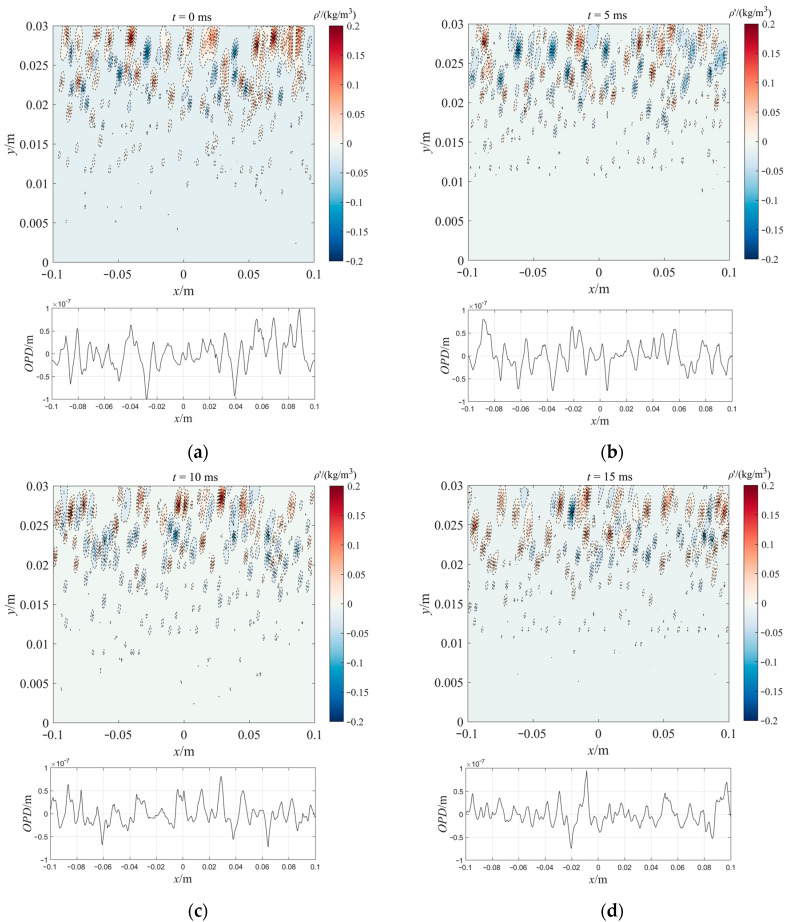
Optical characteristics of XY plane in density proxy model: (**a**) t=0 ms; (**b**) t=5 ms; (**c**) t=10 ms; (**d**) t=15 ms.

**Figure 8 sensors-23-01646-f008:**
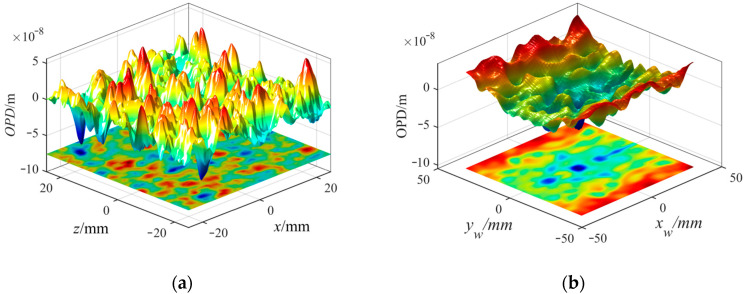
(**a**) OPD of the turbulence density proxy model; (**b**) OPD of the flow field by the CFD.

**Figure 9 sensors-23-01646-f009:**
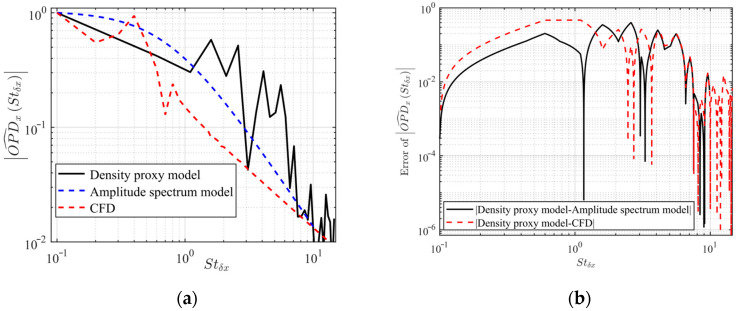
(**a**) OPD spectrum distribution of flow direction; (**b**) OPD spectrum distribution error of flow direction.

**Figure 10 sensors-23-01646-f010:**
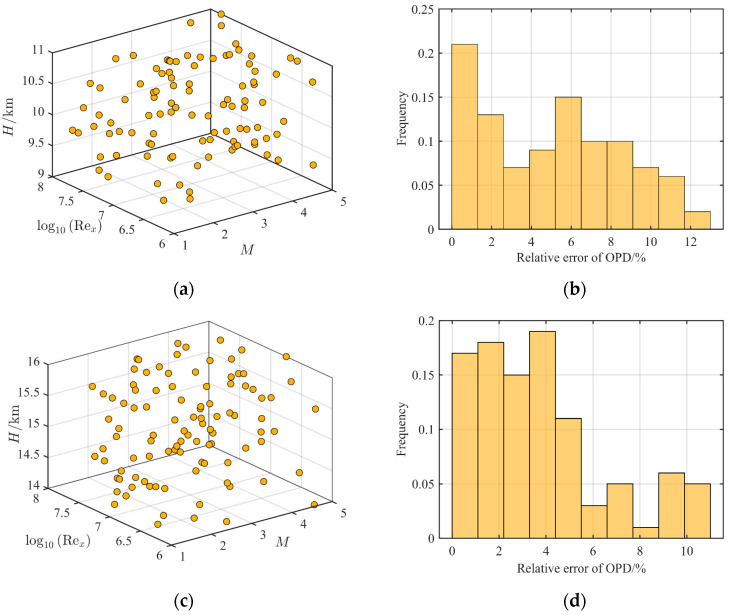
Monte Carlo verification results under different conditions: (**a**) distribution of sampling points at 10 km; (**b**) relative error of OPD at 10 km; (**c**) distribution of sampling points at 15 km; (**d**) relative error of OPD at 15 km.

**Figure 11 sensors-23-01646-f011:**
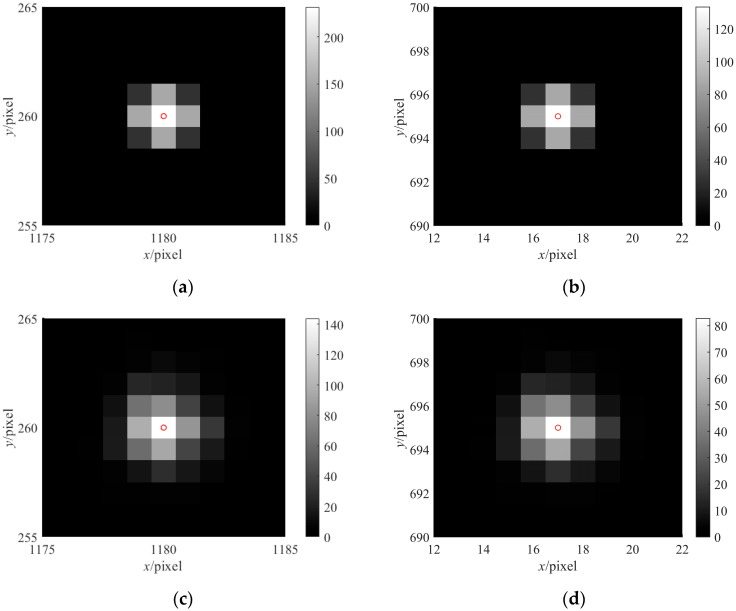
Simulation of distorted star maps: (**a**) star A at 15 km; (**b**) star B at 15 km; (**c**) star A at 5 km; (**d**) star B at 5 km.

**Figure 12 sensors-23-01646-f012:**
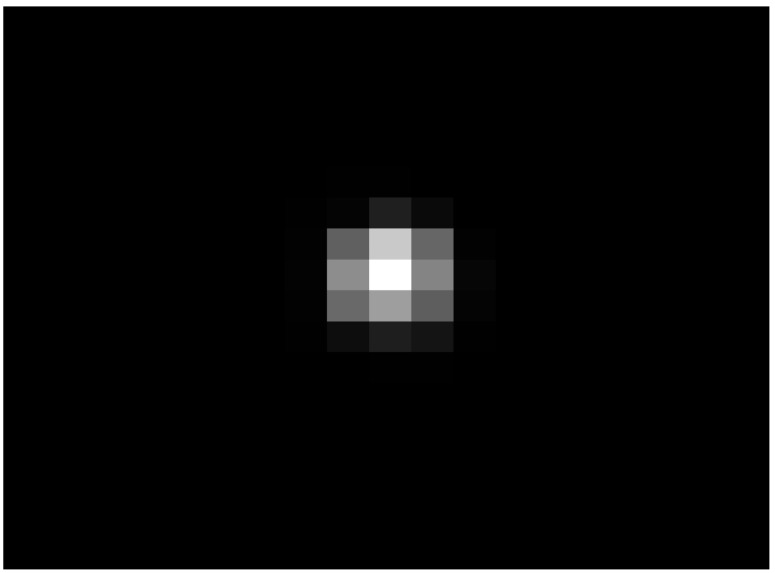
The simulation results of distorted star maps with the previous study [27].

**Table 1 sensors-23-01646-t001:** The gray value at the center of the distorted star maps.

H/km	Center Gray Value of Star A	Center Gray Value of Star B	$SR_{p}$	$SR_{s}$
5	143	83	0.0621	0.0645
10	222	127	0.8613	0.8650

## Data Availability

Data sharing not applicable.

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
