# Peer review of "A Quick Simulation Method for Aero-Optical Effects Based on a Density Proxy Model"

_sensors, 2023, doi:10.3390/s23031646_

Round 1

Reviewer 1 Report

Review of the article "A Quick Simulation Method for Aero-Optical Effects Based on a Density Proxy Model" by Bo Yang, He Yu, Chaofan Liu, Xiang Wei, Zichen Fan and Jun Miao.

The article is written very clearly and understandably. Its main idea is clear and does not require further development. There are the following remarks:

1. Line 102: not a cube, but a cuboid.

2. Lines 141-144: where the values 0.8 and 0.5 come from; lines 149-150: where does the value 0.05 come from?

3. Equation (7): it is necessary to specify characteristic values (range of values) of the k_len parameter.

4. Equation (8): is "c.c" a complex conjugate?

5. Section 2.3: To what extent are the calculations of [27] consistent with the model considered by the authors and the elliptical vortex model from [25]? Why do the coefficients \alpha and \beta have such a different number of significant digits?

6. Lines 185-186: what is the value of 10 km? It is not clear from the text.

7. Figure 4: Change the color and/or linetype for k_\rho=0.01. It is hard to see on the chart.

8. Figure 6: subfigures (a) and (b) cannot be distinguished by eye. Maybe it's worth showing one of them and the error?

9. In the text, the term "height" is used in two different senses: in equation (5) as the distance from the window to the vortex y, and in (11) and (18) as flight altitude H (H_0). It is desirable to correct the text so that these concepts are not confused.

After correcting these remarks, the article may be published.

Reviewer 2 Report

The paper proposes a quick simulation method for aero-optical effects based on a density proxy model (DP-AOQS). The topic is necessary, exciting to the readers, and appropriate for the scope of the sensors journal. The authors have contributed to designing a boundary layer density proxy model to replace the density field in CFD numerical simulation and realize the rapid simulation of aero-optical effects. The paper has a significant novelty. The introduction provides sufficient background.  The research design is appropriate, and the methods are well described. The results are well presented. Overall, the paper is well written. However, some modifications need to be made, and the following comments are considered in a revised version.

1.       It is better to provide the nomenclature section to help the reader.

2.       Please explain the consideration to determine the simulation domain size in Figure 2.

3.       I suggest adding error calculation between the density proxy model and the comparison model (CFD, Amplitude spectrum model) in Figure 9.

4.       It would be very nice to discuss the simulation results of distorted star maps by comparing them with the previous study.

Reviewer 3 Report

Dear authors, enclosed you can find my comments on your work. 

Regards. 

Round 2

Reviewer 3 Report

I am fine with this version

regards